# Ligands and Beyond: Mechanosensitive Adhesion GPCRs

**DOI:** 10.3390/ph15020219

**Published:** 2022-02-11

**Authors:** Hsi-Hsien Lin, Kwai-Fong Ng, Tse-Ching Chen, Wen-Yi Tseng

**Affiliations:** 1Department of Microbiology and Immunology, College of Medicine, Chang Gung University, Taoyuan 33302, Taiwan; 2Department of Anatomic Pathology, Chang Gung Memorial Hospital-Linkou, Taoyuan 33305, Taiwan; nkf362@adm.cgmh.org.tw (K.-F.N.); ctc323@adm.cgmh.org.tw (T.-C.C.); 3Division of Rheumatology, Allergy and Immunology, Chang Gung Memorial Hospital-Keelung, Keelung 20401, Taiwan; 4Department of Medicine, College of Medicine, Chang Gung University, Taoyuan 33302, Taiwan

**Keywords:** adhesion GPCR, GPCR activation, GPS autoproteolysis, mechanotransduction, signaling, tethered ligand

## Abstract

Cells respond to diverse types of mechanical stimuli using a wide range of plasma membrane-associated mechanosensitive receptors to convert extracellular mechanical cues into intracellular signaling. G protein-coupled receptors (GPCRs) represent the largest cell surface protein superfamily that function as versatile sensors for a broad spectrum of bio/chemical messages. In recent years, accumulating evidence has shown that GPCRs can also engage in mechano-transduction. According to the GRAFS classification system of GPCRs, adhesion GPCRs (aGPCRs) constitute the second largest GPCR subfamily with a unique modular protein architecture and post-translational modification that are well adapted for mechanosensory functions. Here, we present a critical review of current evidence on mechanosensitive aGPCRs.

## 1. Introduction

G protein-coupled receptors (GPCRs) form the largest receptor family and function as the major cell communication molecules in response to diverse extracellular stimuli such as photons, odor, and taste [1,2,3]. Among the broad spectrum of ligands that activate GPCRs, mechanical stimulation was recently identified as a new type of activation trigger [4]. Indeed, while ionotropic receptors are considered generally as the prototypical mechano-sensors, several GPCRs are found to be responsive to mechanical force as well [4]. Some, if not all, reported mechanosensitive GPCRs include angiotensin II type 1 receptor (AT1R) [5], GPR68 [6], endothelial histamine H1 receptors (H1Rs) [7], and several adhesion GPCRs (aGPCRs) [8,9,10,11,12,13,14].

Based on phylogenetic characteristics, the GPCR superfamily is further subdivided into five distinct subfamilies including Glutamate, Rhodopsin, Adhesion, Frizzled/Taste2, and Secretin (GRAFS) [15]. In the GRAFS classification system of GPCRs, the 33 different human aGPCR members make up the second largest GPCR subfamily and are thought to be the evolutionary forebear of another GPCR subfamily, the Secretin-like GPCRs [16]. The functional significance of aGPCRs has been amply documented in various physiological and pathological processes such as brain cortex development [17], immune regulation [18,19,20], fertility [21], and tumorigenesis [22,23]. While it is beyond the scope of the present review to discuss the functional roles of aGPCRs (please see [24,25] for detail), the underlying mechanisms for the diverse functions of aGPCRs are mainly due to their complex structural organization and diverse receptor activities that integrate extracellular adhesion and intracellular signaling [26]. Consequently, one of the major efforts in the aGPCR research field is the identification of their cognate endogenous and exogenous ligand molecules that trigger aGPCR activation. Nevertheless, several aGPCRs are now found to be activated by mechanical stimulation in the absence or presence of specific ligands [8,9,10,11,12,13,14]. In this review article, we first discuss the structural characteristics and activation mechanisms of aGPCRs to explain why they are suited for mechanosensing. Finally, we summarize the current understandings of specific aGPCRs that function as potential mechanosensitive receptors.

## 2. Adhesion GPCRs: Structural Characteristics and Activation Mechanisms

Unlike the canonical GPCRs, the structural feature of aGPCRs is hallmarked by a notably extended (ranging from ~300 to 6000 amino acids) extracellular region (ECR) proximal to the seven-span transmembrane (7TM) region [24,27]. Intriguingly, the N-terminal region of aGPCR-ECR usually contains various characteristic cell-adhesive protein motifs arranged either in tandem repeats or in different combinations. Indeed, several well-known protein motifs including the epidermal growth factor-like (EGF), immunoglobulin-like (Ig), lectin-like, pentraxin (PTX), thrombospondin (TSP), and leucine-rich repeat (LRR) domains are identified in the aGPCR-ECR that are involved in ligand-binding and cellular adhesion [24] (Figure 1A).

A characteristic GPCR autoproteolysis inducing (GAIN) domain is usually located immediately downstream of the cell-adhesive protein motifs [24,28] (Figure 1A). The GAIN domain is an evolutionarily conserved protein configuration minimally required and self-sufficient to initiate a novel post-translational auto-proteolytic process [28,31]. Autoproteolysis takes place at a highly conserved HL/T(S) sequence of the GPCR proteolysis site (GPS) via a series of nucleophilic reactions [32,33]. This leads to the precise dissection of the full-length receptor and the generation of an N-terminal fragment (NTF) and a C-terminal fragment (CTF) that form a tightly-associated non-covalent dual-subunit complex on the cell surface [24]. The GPS motif is uniquely present in aGPCRs and very few other cell surface receptors such as polycystin-1 [33]. In general, GPS autoproteolysis is considered an essential process for the maturation and function of the majority of aGPCRs [34]. As a result, most mature aGPCRs are expressed as a bipartite receptor molecule consisting of a cell-adhesive NTF and a signaling 7TM-containing CTF [24] (Figure 1A).

Multiple aGPCR activation mechanisms have been proposed following the identification of the ligands/binding partners of various aGPCRs and the realization of their dual-subunit composition [35,36,37]. Among these different potential activation mechanisms, the novel tethered agonism model is most relevant to the idea that aGPCRs could function as mechanosensitive receptors [29,30] (Figure 1B). As described, autoproteolysis at the GPS motif would generate a new short N-terminal sequence (usually starting at Thr or Ser) of the CTF (also called the *Stachel* sequence), which is usually embedded in and closely surrounded by the rest of the GAIN domain [28]. Surprisingly, however, it has been demonstrated that strong aGPCR activities are readily induced in the CTF-only (NTF-less) form or by the addition of free *Stachel* peptides [29,30,37,38]. This receptor activation mechanism is strikingly similar to that of another GPCR family, the protease-activated receptors (PARs), that are activated by a newly exposed tethered peptide ligand following the proteolytic cleavage and removal of their N-terminal sequence [39]. Hence, the aGPCR tethered agonism mechanism stipulated that the NTF-CTF complex is dissociated or dislocated upon the binding of specific ligands/binding partners, allowing the unmasking of the *Stachel* peptide to bind and activate the 7TM region of its own CTF, most likely due to conformational changes [40,41]. More specifically, the shedding/separation of NTF from CTF is believed to be the prerequisite step for the activation of aGPCR-CTF by the tethered *Stachel* peptide (Figure 1B).

## 3. Mechanosensitive Adhesion GPCRs

Due to the unique GPS autoproteolysis and the unusual NTF-CTF organization as well as the fact that most aGPCR ligands identified to date are cell surface and/or extracellular matrix (ECM) proteins, it is highly suggestive that some aGPCRs are prone to respond to mechanical disturbance during cell-cell or cell-ECM interaction (Figure 2). Indeed, based on the tethered agonism model described above, mechanical force is one of the conspicuous cues for NTF shedding and it seems that the GAIN domain-containing receptors have well-fitted molecular apparatuses for sensing mechanical stimulation in the extracellular microenvironments. It is well known that polycystin-1 (PC1) interacts with polycystin-2 (PC2) ion channels to mediate fluid flow sensing in the renal epithelium [42,43]. PC1 is a 11TM receptor closely linked to the pathogenesis of autosomal dominant polycystic kidney disease (ADPKD) and GPS cleavage plays an essential role in the structural stability, intracellular trafficking, and cellular function of PC1 [44,45,46]. Considering the role of GPS proteolysis in the structural-functional importance of PC1, it is not surprising that some aGPCRs have also been identified as possible mechanosensing receptors. Notwithstanding, GPS autoproteolysis does not occur in all aGPCRs due to the lack of a complete GAIN domain or the absence of a consensus GPS sequence [24,47,48,49]. Consequently, functional activation of aGPCRs includes both GPS proteolysis-dependent and GPS proteolysis-independent mechanisms [36,37]. Of great interest, the latest studies have indicated that the tethered agonism model is also applicable to the activation of uncleaved aGPCRs and that certain mechanosensitive aGPCRs may indeed function via a GPS cleavage-independent mechanism [8,50] (Figure 2).

In recent years, increasing evidence for a mechanosensing role of aGPCRs has been substantiated by in vitro and in vivo studies as well as the mechanical mechanisms of novel clinical diseases [4,8,51]. Here, seven potential mechanosensitive aGPCRs including ADGRE2/EMR2, ADGRE5/CD97, ADGRL1/Latrophilin/CIRL, ADGRV1/VLGR1, ADGRG1/GPR56, ADGRG5/GPR114, and ADGRG6/GPR126 are discussed (Table 1).

### 3.1. ADGRE2/EMR2

ADGRE2/EMR2 is a member of the ADGRE subgroup of aGPCRs that was also called the EGF-TM7 receptors due to the presence of tandem repeats of EGF domains in their ECR [53,54,55]. EMR2 is a fully GPS-processed aGPCR that was used previously to define the novel GPS auto-proteolytic mechanism [32]. Interestingly, EMR2 is a human myeloid cell-restricted receptor and is expressed differentially in monocyte, macrophage, granulocyte, and myeloid dendritic cell [56,57,58]. Up-regulated surface EMR2 expression on blood neutrophils is positively associated with the severity and overall mortality of systemic inflammatory response syndrome (SIRS) and liver cirrhosis patients [59,60]. Furthermore, EMR2 has been shown to be involved in the activation, adhesion, and migration of myeloid cells, while dermatan sulphate (DS) was identified as its endogenous glycosaminoglycan ligand [61,62,63]. It was shown subsequently that the interaction of DS and EMR2 on myeloid cells might facilitate the inflammatory recruitment of monocytes into the synovium of rheumatoid arthritis (RA) patients [64].

Importantly, Boyden et al. analyzed families of patients with autosomal dominant vibratory urticaria (VU) and identified a missense EMR2-C492Y variant as the sole cause of the rare autoinflammatory disorder [9]. Clinical manifestations of VU include localized hives resulting from a dermal allergic response to repetitive mechanical stimulation such as dermal vibration [9]. EMR2 is expressed in mast cells and the separation of EMR2-NTF from EMR2-CTF induced rapid mast cell activation and histamine discharge, hence the urticaria. It was demonstrated that the change of Cys_492_, located within the GPS motif, to Tyr, greatly destabilizes the EMR2 NTF-CTF interaction rendering the receptor complex highly prone to be separated upon vibratory stimulation. Thus, vibration-facilitated EMR2-NTF shedding is more pronounced in mast cells expressing the EMR2-C492Y variant than those expressing the wild-type EMR2. Critically, EMR2-induced mast cell degranulation is GPS cleavage- and vibration-dependent, while the vibration-induced EMR2-NTF shedding requires the binding of DS or an EMR2-GAIN domain-specific mAb immobilized on culture plates [9]. In conclusion, these results clearly support a ligand-dependent mechano-sensing function of EMR2 in mast cells (Figure 3A).

It is not known whether similar mechanosensitive functions of EMR2 are also identified in monocytes/macrophages and neutrophils of VU patients. Nevertheless, we have shown recently that EMR2 activation in macrophages promotes cellular differentiation and inflammatory activation as well as the induction of the NLRP3 inflammasome activation (2nd) signal [19,65]. Of interest, these EMR2-induced macrophage activation phenotypes were observed only when the receptor was activated by an immobilized GAIN domain-specific mAb. The free, soluble form of the same mAb has not any stimulatory effect [19,65]. More recently, Irmscher et al. identified an EMR2-specific serum ligand, the complement Factor H-related protein 1 (FHR1), that also functions as a molecular sensor of necrotic cells [66]. Interestingly, it was shown that the binding of FHR-1 to EMR2 on monocytes induced swift NLRP3 inflammasome activation, but only in the presence of normal human serum and when FHR1 was immobilized [66]. While remained unconfirmed, the independent realization of the critical importance of ligand immobilization for EMR2-specific activation in monocytes/macrophages suggested strongly that some form of mechanical force such as the possible molecular pulling force caused by receptor cross-linking through immobilized multivalent ligands is likely involved. As monocytes/macrophages are normally subjected to various shear forces and ECM compression/pressure in the blood stream and interstitial tissues, it will be of great interest in the future to investigate possible EMR2-NTF shedding of monocytes/macrophages and its functional outcomes. In addition to the dissociation of the EMR2 NTF-CTF complex, it will be equally interesting to understand the effect of the mechanical force on the binding of EMR2 and its cellular ligand(s) such as DS that is of a low binding affinity. Overall, EMR2 is an archetypical mechanosensitive aGPCR that requires ligand interaction and mechanical stimulation for receptor activation in a GPS cleavage-dependent fashion.

### 3.2. ADGRE5/CD97

ADGRE5/CD97, another ADGRE subfamily member, partakes a high degree of structural similarity with EMR2. In fact, the sequences of the five tandem EGF domains of both receptors are 97.5% identical, while their GAIN domains and 7TM regions share a ~50% sequence homology [55]. Unlike EMR2 however, CD97 is expressed more broadly in cells of the hematopoietic system such as monocytes, neutrophils, and T cells. In addition, CD97 is also highly expressed in epithelial cells, muscle cells, and many different types of cancer cells [56]. Several cellular ligands of CD97 including the decay-accelerating factor (DAF/CD55) [67], dermatan sulphate (DS) [68], α_5_β_1_ and α_v_β_3_ integrins [69], and Thy-1 (CD90) [70] have been identified. As a result, CD97 has been implicated functionally in innate immune defense, T cell activation, auto-inflammatory diseases such as rheumatoid arthritis and multiple sclerosis, as well as tumor angiogenesis and development [56,69,71,72,73,74].

In the immune system, CD97 is known to play a role in granulocyte migration and homeostasis as well as the regulation of T cell activation [73,75,76]. Importantly, these CD97-mediated immune functions are partly dependent on its interaction with CD55, which is a glycosylphosphatidylinositol (GPI)-anchored complement regulatory protein [67]. Interestingly, while studying the in vivo and in vitro functional significance of CD97-CD55 interaction, Karpus et al. unexpectedly found that contact of circulating leukocytes with CD55 results in a rapid CD97 downregulation on the cell surface [10]. Indeed, increased surface CD97 levels were detected in CD55-deficient leukocytes when compared to cells from the wild-type mice. However, the enhanced CD97 expression on CD55-deficient leukocytes quickly returned to the normal levels following their adaptive transfer into wild-type mice, obviously because of the interaction of CD97 with CD55. Conversely, adaptive transfer of the wild-type leukocytes into the CD55-deficient mice lead to a significantly increased surface CD97 expression. Intriguingly, CD97 downregulation in vivo occurred within minutes after cellular contact with CD55 and increased levels of soluble CD97 were concomitantly detected in mouse plasma, apparently due to CD97-NTF shedding. Subsequent in vitro cell-coculture studies showed that shear stress is absolutely required for CD97 downregulation on CD55-deficient leukocytes. This was further confirmed by in vivo experiments in which intact blood circulation is shown to be essential for CD55-mediated downregulation of CD97 [10]. The authors concluded that physical forces such as shear stress in the blood stream are necessary for the shedding of CD97-NTF, hence the downregulation of CD97 surface levels on circulating leukocytes by CD55 (Figure 3B). Therefore, CD97, like EMR2, seems to display a ligand-dependent mechano-sensing function in leukocytes.

To provide a structural basis of the specific CD97-CD55 interaction, Niu et al. generated a novel chimeric protein complex containing the EGF-1, 2, 5 domains of CD97 and the short consensus repeat (SCR)-1, 2, 3, 4 domains of CD55 interlinked by a short flexible sequence [77]. Structural determination of this chimeric protein by crystallography revealed an overall antiparallel binding configuration involving the SCR1–3 domains and all three EGF domains. Specifically, three major domain-binding interfaces, namely the EGF1–SCR2/SCR3 interface, the EGF2–SCR1/SCR2 interface, and the EGF5–SCR1 interface, are involved to stabilize the CD97-CD55 complex. Furthermore, biophysical analysis of the CD97-CD55 interaction by surface plasmon resonance detected a binding K_D_ of 3.2 μM with a slow on-rate (K_on_) of 546 M^−1^S^−1^ and off-rate (K_off_) of 1.73 × 10^−3^S^−1^ [77]. Overall, these results indicate that the CD97-CD55 complex adapts a force-resistant shearing stretch geometry hence providing a molecular basis for the mechanosensing function of the CD97-CD55 receptor-ligand pair.

To study how CD97 responds to mechanical stimuli, Hilbig et al. examined the biochemical phenotypes of CD97 following mechanical disturbance of cells [75]. A rapid phosphorylation of the intracellular PDZ-binding motif (PBM) of CD97 was identified at S740 in shear-stressed cells and this modification caused the disruption of CD97 interaction with the intracellular scaffold protein DLG1. The mechanical force-induced CD97 phosphorylation further leads to cellular detachment. The critical importance of PBM phosphorylation for the CD97-dependent mechanoresponse was verified by experiments using cells expressing PBM-deleted CD97 and CD97-knockout. Interestingly, however, CD97-NTF shedding was not observed in these experimental settings [75]. This is probably because cell samples used in this study are mostly adherent fibrosarcoma, breast cancer, and colorectal cancer cell lines. Furthermore, no involvement of specific CD97-ligand was indicated. Unlike what was found in circulating leukocytes, the authors suggested alternatively that CD97 may be an integral component of a cellular mechanosensitive receptor complex in which its PBM-mediated signaling is actively involved in the intracellular transmission of mechanical stimuli [75]. In other words, CD97 may likely be involved in mechano-transduction instead of a direct “sensor” in certain adherent cell types.

In recent years, mechanical force has been shown to play important roles in the interaction and function of various immunoreceptor-ligand pairs, highlighting the significance of the mechanosensing properties of immunoreceptors [76]. Interestingly, CD97 was found recently to stabilize the immunological synapse between dendritic cells and T cells, which was known to be subjected to various molecular forces before and after the formation of the T cell receptor (TCR) and antigenic peptide-major histocompatibility complex (pMHC) conjugate [78]. In addition, Capasso et al. demonstrated that CD97-CD55 interaction might also exert a functional effect via CD55 for the costimulation and activation of CD4^+^ T cells [20]. Therefore, it will be of importance in future studies to investigate the possible bidirectional effects of mechanical force on the effector functions of CD97-expressing cells and CD97-ligand expressing cells during immune cell interaction. The influence of mechanical force on the interaction of CD97 with its other ligands in different cell types is also of great interest.

### 3.3. ADGRG1/GPR56

ADGRG1/GPR56 belongs to the ADGRG subgroup and contains a novel Pentraxin/Laminin/neurexin sex-hormone-binding globulin-like (PLL) domain in front of the GAIN domain [79]. GPR56 was one of the first aGPCRs to be deorphanized and its multiple ligands/binding partners, including CD9/CD81 [80], tissue transglutaminase (also named transglutaminase-2, TG2) [81], collagen III [82], laminin [83], heparin/heparan sulfate [84], progastrin [85], the essential amino acid L-Phe [86], and phosphatidylserine (PS) [87] have been identified over the years. Stoveken et al. investigated the activation mechanism of GPR56 and ADGRF1/GPR110 and independently established the tethered agonism model of aGPCR activation [30]. The authors showed that treatment of cell membranes with urea could dissociate the NTF-CTF complex of both receptors, subsequently leading to receptor activation. The conserved N-terminal sequences of the CTF were found to be critical for receptor activation and screening of a panel of different lengths of the N-terminal peptides identified a 7-residue peptide and a 12-residue peptide capable of inducing optimal GPR56 and GPR110 activation, respectively [30].

GPR56 transcripts were broadly detected in the brain, thyroid, pancreas, kidney, testis, skeletal muscle, and the hematopoietic system [88]. Consistently, GPR56 has been found to be functionally important in diverse physiological and pathological processes [88]. The most recognized biological function of GPR56 is its essential role in the normal cerebral cortical development as its loss-off-function resulted in bilateral frontoparietal polymicrogyria (BFPP), a rare genetic disorder affecting the cerebral cortex, in the frontal and parietal lobes of the brain [17]. The interaction of GPR56 with collagen III, its ECM ligand in the brain, through the PLL domain plays a critical role in the disease mechanisms of BFPP [82]. Interestingly, physical forces were thought to be involved in various neuronal developmental processes including cortical folding, which is closely linked to the cortical malformation phenotypes caused by the BFPP-associated GPR56 defects [89].

Luo et al. found that collagen III binding reduced the surface levels of GPR56-NTF and triggered the translocation of GPR56-CTF into the lipid raft microdomains and RhoA signaling [90]. Later experiments by Zhu et al. showed that the collagen III-induced GPR56 activation absolutely depended on efficient GPS proteolysis and dissociation of NTF from CTF [51]. While it is not known whether any kind of physical force is involved in collagen III-induced GPR56 activation, it is of interest to note that mature collagen III monomers normally form a triple-helix conformation that further assembles into macromolecular super-helical fibrils [91]. This means that multiple GPR56-binding sites are available in collagen III and hence the possible cross-linking/pulling effects on the receptor molecules. Likewise, our investigation of GPR56 activation in human melanoma cell lines by specific GPR56 mAbs also indicated that Ab-induced GPR56 activation resulted in NTF shedding and the *Stachel* peptide-dependent CTF-mediated Gα_12/13_-RhoA signaling [92]. The Ab-induced GPR56 activation was GPS cleavage-dependent and totally required Ab immobilization, likely again for the ligation/cross-linking of receptors.

In blood hemostasis, exposed subendothelial collagen III in injured blood vessels is known to facilitate platelet aggregation that plays a critical role in blood clotting. Interestingly, Yeung et al. recently showed that GPR56 expressed in platelets is involved in the shear force-dependent adhesion and activation of platelets to immobilized collagen [12]. Again, stimulation of GPR56 receptor by immobilized collagen under shear force led to NTF shedding and platelet shape changes due to CTF-mediated Gα_13_ signaling in a *Stachel* peptide-dependent manner (Figure 3C). In sum, GPR56 is a novel platelet receptor with a dual collagen-responsive and shear force-sensing function.

In addition to the central nervous system (CNS), GPR56 is also involved in the myelination of the peripheral nervous system (PNS), myelin repair in CNS neurons, the proliferation of oligodendrocyte precursor cells (OPCs), and mechanical overload-induced muscle hypertrophy [88,93]. Giera et al. showed that GPR56 is expressed in OPCs and its activation by microglia-derived TG2 in the presence of the ECM laminin protein promotes OPC proliferation during developmental myelination and improves remyelination in two animal models of myelin repair [83]. Interestingly, receptor signaling analyses in an in vitro cell-based system showed that GPS cleavage and laminin are absolutely required for TG2-induced GPR56 activation [51]. More importantly, GPR56 activation and signaling is dependent on the shedding of GPR56-NTF following the tripartite GPR56-TG2-laminin interaction. Similar to collagen molecules, laminins are high molecular weight heterotrimeric ECM proteins that can further polymerize into large aggregates [94]. On the other hand, TG2 works to crosslink proteins by forming proteolysis-resistant inter- and/or intramolecular bonds [95]. It is likely that the binding and ligation of GPR56 by the TG2/laminin complex exerts a tractive force to dissociate its NTF from CTF.

In skeletal muscle, up-regulated GPR56 expression was induced by resistance exercise. GPR56 was identified as a transcriptional target of the peroxisome proliferator-activated receptor gamma coactivator 1-alpha 4 (PGC-1α4) isoform that is known to promote muscle hypertrophy [93]. Furthermore, Gpr56 knockdown in murine muscle cells reduced PGC-1α4-induced muscle hypertrophy. Conversely, GPR56 overexpression resulted in myotube hypertrophy. Up-regulated expression of Gpr56 and collagen III was identified in an animal model of overload-induced muscle hypertrophy, which was attenuated in Gpr56-deficient mice. These results denote a role for collagen III-induced Gpr56 signaling in the response of skeletal muscle to a mechanical tension. Taken together, these results suggested that GPR56 is activated by the tethered *Stachel* peptide exposed after NTF shedding, possibly caused by the mechanical force of receptor cross-linking by collagen III, immobilized mAb, and TG2/laminin either in the absence or presence of additional shear forces (Figure 3C).

### 3.4. ADGRG5/GPR114

ADGRG5/GPR114 is encoded by the *ADGRG5* gene located on the human chromosome 16q21, right next to the *ADGRG1* and *ADGHG3* genes which encode GPR56 and GPR97, respectively [96]. This suggests a close evolutionary relationship among the three ADGRG molecules. Indeed, similar to GPR56 and GPR97, GPR114 transcripts were detected in cells of the hematopoietic system such as human eosinophils and mouse lymphocytes, monocytes, macrophages, and dendritic cells [24]. In addition, GPR114 mRNA expression was also found in the colon, spleen, and thymus. Based on the sequence analysis, GPR114 is one of the smaller aGPCRs with a shortened GAIN domain and an undefined N-terminal segment. Little is known of GPR114 protein expression characteristics and its interacting ligands. However, 3-*α*-acetoxydihydrodeoxygedunin (3-*α*-DOG) and dihydromunduletone (DHM) were identified recently as a small molecule partial agonist and a novel antagonist for both GPR114 and GPR56, respectively, probably because the two aGPCRs shared almost identical tethered peptide sequences [97,98]. This suggests that GPR114 likely could also be activated by its *Stachel* tethered peptide.

Indeed, Wilde et al. identified and compared 2 naturally-occurring mouse Gpr114 isoforms that differed only in one residue (Q230) of the *Stachel* sequence due to alternative splicing [11]. An in vitro cell-based cAMP signaling assay was performed to show that both Gpr114 isoforms were able to respond to the exogenously added *Stachel* peptides. Interestingly, however, in comparison to the ΔQ230 isoform, the full-length Gpr114 variant displayed a significantly higher basal receptor activity which was even more augmented by mechanical shaking. Surprisingly, no known Gpr114-specific ligand was involved in the receptor activity assay and Gpr114 was found to be an uncleaved aGPCR [11]. The authors concluded that the N-terminal half (the first 6–7 core residues) of the *Stachel* sequence is critically responsible for its agonistic activity and the C-terminal half (starting from the 8th residue) of the *Stachel* sequence functions mainly to orientate the agonistic core sequence toward the 7TM domain [11]. It the case of a GPS-unprocessed constitutive active aGPCR such as Gpr114, it is suggested that the core agonistic *Stachel* sequence is somewhat unconcealed and prebound to the 7TM to induce basal receptor activities. A mechanical stimulation such as vibratory shaking likely results in a further conformational change and full exposure and extensive interaction of the *Stachel* peptide with the 7TM region eventually leading to the maximum receptor activity (Figure 3D). In summary, some aGPCRs may respond to mechanical stimuli in a ligand-independent and GPS cleavage-independent, yet the *Stachel* peptide-dependent manner.

### 3.5. ADGRG6/GPR126

ADGRG6/GPR126 is an unusual ADGRG subgroup member that consists of a relatively long ECR of well-defined protein domains, including the CUB (Complement C1r/C1s, Uegf, Bmp1), PTX (Pentraxin), SEA (Sperm protein, Enterokinase and Agrin), and HormR (Hormone Receptor) motifs [99]. Like GPR56, GPR126 was one of the first few aGPCRs employed to establish the tethered agonism mechanism of aGPCR activation [29]. GPR126 is ubiquitously expressed in many tissues/organs, but significant GPR126 expression is detected in specific cell types that are exposed to shear stress or mechanical stimulation such as endocardium and arterial endothelial cells, chondrocytes, and bladder umbrella cells, suggesting a role in the mechanoresponse [100]. To date, three different cellular ligands, namely collagen-IV [101], laminin-211 [52], and prion protein PrP^C^ [102] have been identified to bind to GPR126-ECR.

The Gpr126-deficient mice show an embryonic lethal phenotype mainly due to cardiac abnormality [103]. In addition, Gpr126 deficiency also leads to multiple defects in peripheral nerves [104], inner ear (in zebrafish) [105], placental development [106], and lack of myelination in the peripheral nervous system (PNS) [107]. More recent studies in cell type-specific Gpr126 knock out animals further indicated a role for GPR126 in the regulation of body length and bone mass as well as the maintenance of spinal alignment [108,109]. In line with these, GPR126 mutations and/or genetic variants in humans are associated with several disorders, including adolescent idiopathic scoliosis [110], arthrogryposis multiplex congenita [111], and intellectual disability [112].

Interestingly, it was found that Gpr126 might exert its organ-specific physiological functions via a domain-dependent mechanism. As such, Gpr126-NTF itself alone seems to be critical for heart development independently of the CTF. By contrast, Gpr126-CTF is essential for the normal development of PNS [103]. Myelination in PNS is initiated by Schwann cells and it has been shown that GPR126 activation in Schwann cells leads to elevated cAMP levels via Gαs-coupled signaling and the formation of myelin sheath [107,110,113]. Importantly, an intact *Stachel* peptide was found to be essential for the GPR126-mediated signaling and cAMP elevation [29]. Consistent with its domain-dependent organ-specific functions, it was shown that the GPR126-NTF is necessary and sufficient for axon sorting while the GPR126-CTF is involved in myelin wrapping during Schwann cell development [52]. Interestingly, the heterotrimeric laminin-211 ECM protein is also found to be required for the development and myelination of Schwann cells [114]. Later studies further demonstrated that overexpression of a laminin-211 subunit rescues the myelination phenotype of the gpr126 hypomorphic zebrafish mutants [52]. These results collectively support the important role of laminin-211-induced GPR126 signaling in Schwann cell development and myelination.

Direct binding of GPR126 by collagen-IV and PrP^C^ resulted in increased cAMP accumulation in in vitro cell-based systems. Unexpectedly, however, the interaction of GPR126 with laminin-211 led to reduced cAMP levels in static conditions, but enhanced cAMP accumulation following mechanical challenge (vibration or shaking) [52]. This indicated potential structural changes of GPR126-NTF and the induction of the *Stachel* peptide-mediated receptor activation in the presence of laminin-211 and vibratory stimulation. As laminin-211 polymerization has been known to play an essential role for Schwann cell development in mice and a non-polymerizable mutant laminin-211 failed to rescue the myelination defects in gpr126 hypomorphic zebrafish mutants, it is believed that laminin-211 polymerization is partly the physiological source of the mechanical force that stimulated GPR126 activation in vivo [52] (Figure 3E).

Results of the latest structural-functional study of zebrafish gpr126 revealed a diverse range of flexible conformations of GPR126-ECR that showed differential receptor activities [99]. Specifically, a closed ECR conformation was identified for the alternatively-spliced gpr126-S2 isoform in which the CUB domain interacted closely with the HormR domain that was stabilized by a Ca^2+^ binding site in the CUB domain and a disulfide-linked loop between the SEA and PTX domains. Interestingly, the addition of a 23-a.a. sequence between the SEA and PTX domains in the gpr126-S1 isoform disrupted the stable CUB-HormR interaction and allowed the adaptation of dynamic open-like conformations. Importantly, cells expressing the highly mobile S1 isoform displayed an increased basal receptor activity than those expressing the S2 isoform [99]. These results clearly indicate that conformational changes of GPR126-ECR play a role in its receptor activation and signaling. As these structural studies were done in the absence of any GPR126-ligands, it is conceivable that a more profound effect on the ECR conformations and receptor activities will be exerted upon ligand binding in the absence or presence of mechanical force. Altogether, GPR26 is a *Stachel*-peptide dependent mechanosensitive aGPCR expressed in cell types responding to constant mechanical challenge.

### 3.6. ADGRL1/LPHN1/CIRL

ADGRL1/LPHN1 was also named latrophilin-1 or CIRL as it was identified originally as the calcium-independent receptor for α-latrotoxin [113,115,116]. Four aGPCRs are included in the ADGRL class with three members (LPHN1, LPHN2, and LPHN3) sharing a similar N-terminal structure of ECR, which contains a rhamnose-binding lectin (RBL) domain, an olfactomedin-like domain, and a HormR motif [24]. The crystal structure of the rat LPHN1 GAIN domain was solved by Araç et al. to establish it as an evolutionarily ancient protein fold both necessary and sufficient for GPS autoproteolysis. It is believed that a functional GAIN domain provides a proper chemical environment to catalyze the autoproteolytic reaction at the GPS motif [28].

Consistent with its ability to bind α-latrotoxin, which is a spider venom toxin with a potent presynaptic function to release neurotransmitters from sensory and motor neurons, latrophilins are highly expressed in the nervous system. Resultantly, latrophilins have been associated with diverse neurological disorders such as autism spectrum disorder (ASD), bipolar disorder (BD), attention deficit and hyperactivity disorder (ADHD) and substance use disorder (SUD) [117]. In addition to the exogenous α-latrotoxin ligand, latrophilins have been shown to bind many endogenous cellular ligands including teneurins, neurexins, the fibronectin and Leucine-rich transmembrane proteins (FLRTs), and contactins either in a *cis*- or *trans*-mode [117].

Interestingly, functional studies in *Drosophila* showed that dCirl is expressed in the dendrites and cilia of chordotonal neurons and it is needed for coordinated locomotion of larvae [8]. Specifically, dCirl sensitizes chordotonal sensory neurons in a cell-autonomous manner for the perception of tactile, proprioceptive, and auditory stimuli by reducing intracellular cAMP levels. dCirl interacts with the TRP channels to modulate the ionotropic receptor currents in an intact *Stachel* peptide-dependent but GPS cleavage-independent fashion (Figure 3F) [14]. Of interest, the length of the dCIRL NTF was found to modulate the mechanosensitivity of chordotonal neurons, suggesting that the response of dCIRL to mechanical challenge is regulated by the size and malleable property of its ECR [118].

In addition to the enhancing effect on the low-threshold mechanosensory neurons responsible for gentle touch, sound, and proprioceptive input, Dannhäuser et al. recently showed that dCirl is also expressed in high-threshold mechanical neurons that respond to strong mechanical stimuli [119]. Interestingly, however, dCirl exerts an opposite effect on the nociceptive neurons by dampening their response to mechanical stimulation. Consistently, reduced Cirl1 expression was detected in rat nociceptors during allodynia. In direct contrast to the situation in touch-sensitive chordotonal neurons, an intact *Stachel* sequence is dispensable for the antinociceptive effect of dCirl. Therefore, CIRL is able to execute completely opposite effects on low-threshold mechanosensors and high-threshold nociceptors by down-regulating cAMP levels a *Stachel*-dependent and *Stachel*-independent mechanism, respectively [119]. In conclusion, CIRL is a mechanosensitive aGPCR capable of bidirectional regulation of distinct mechanosensory modalities of different physiological messages. At present, it is not clear about the role of ligands in the mechanosensory functions of dCirl in fly, but mammalian latrophilins are found to interact with various cell surface protein ligands to establish neuronal connections at the pre- and post-synaptic compartments. On the other hand, latrophilins interact with distinct scaffolding proteins intracellularly to modulate cytoskeletal arrangement and cellular adhesion [117]. In the future, it will be worthwhile to investigate the nature and role of potential mechanical force in regulating these latrophilin-mediated cellular functions in the mammalian nervous system.

### 3.7. ADGRV1/VLGR1

ADGRV1/VLGR1 is the largest known cell surface protein that contains a total of ~6300 amino acids, with multiple copies of calnexin (Calx)-β domain, a pentraxin domain, and epilepsy-associated epitempin (EPTP) repeats in the ECR [120,121]. VLGR1 is highly expressed in the developing CNS and eye, especially associated with the optic nerve [121]. Multiple alternatively spliced VLGR1 isoforms have been identified and VLGR1 mutations in humans have been associated with febrile and afebrile seizures [122] and the Usher’s syndrome (USH), a severe sensory-neuronal disorder that affects both vision and hearing [123]. Likewise, Vlgr1-deficient mice are susceptible to audiogenic seizure and epilepsy, essentially phenocopying USH [124,125].

In retinal photoreceptor cells and auditory hair cells, VLGR1 seems to be an intrinsic constituent of adhesion fibrous linkers between neighboring membranes that are ideally located for sensing mechanical changes [126,127]. Interestingly, VLGR1 forms an extensive protein interactome with other USH proteins in both cell types [128,129]. A recent systematic affinity proteomics analysis of the VLGR1 protein interactome by Kusuluri et al. identified several components of focal adhesion (FA), which is a large protein complex associated with the integrin-ECM interaction [13,130]. The authors further demonstrated experimentally that VLGR1 is indeed an integral component of FAs and VLGR1 deficiency reduces the number and length of FAs, impedes cell spreading and migration as well as a cellular response to mechanical stretch (Figure 3G) [13]. Thus, VLGR1 in FAs seems to function as a metabotropic mechanoreceptor at the cell-ECM interface. These results indicated that VLGR1 might play a role in cellular spreading and migration by regulating the bidirectional signaling of the integrin-FA complex, namely the “outside-in” signal such as the shear forces induced by the cell-ECM interaction and the “inside-out” signal transmission of intracellular forces resulted from cytoskeletal changes [131,132]. Considering its critical importance in epileptic seizures, it will be important next to investigate the possible mechanosensing role of VLGR1 in the disease mechanism of epilepsy.

## 4. Conclusions and Future Perspectives

The unusual NTF-CTF bipartite design coupled with the dual cell-adhesive and signaling functions have made aGPCRs credible biosensors for mechano-transduction. In addition, the cell-type specific expression characteristics of some aGPCRs such as EMR2, CD97, and GPR56 in circulating leukocytes, VLGR1 in auditory hair cells, and LPHN1 in mechanosensory neurons further highlighted the role of aGPCRs in mechano-sensing. Indeed, with more than one-fifth (7 out of 33) of the aGPCR subfamily already being found to function as potential mechanosensitive receptors, it is reasonable to suggest that mechano-stimulation is likely a dominant mode of aGPCR activation mechanism. As cells may be subjected to a multitude of mechanical factors such as shear stress, the traction force during cell adhesion, rolling, and migration, differential tissue rigidity/stiffness, and ECM compression, a future investigation will need to focus on the nature and type of mechanic forces needed to trigger the activation of specific aGPCRs. The dependence of specific cellular ligands and the role of GPS autoproteolysis and the *Stachel* peptide for the mechanosensitive function of aGPCRs are also important questions. Finally, understanding the signaling events and functional outcomes of mechano-activated aGPCRs will not only reveal their various physio-pathological significance such as VU and CNS myelination, but also facilitate the development of possible therapeutic agents and drug candidates for mechano-sensory system disorders.

## Figures and Tables

**Figure 1 pharmaceuticals-15-00219-f001:**
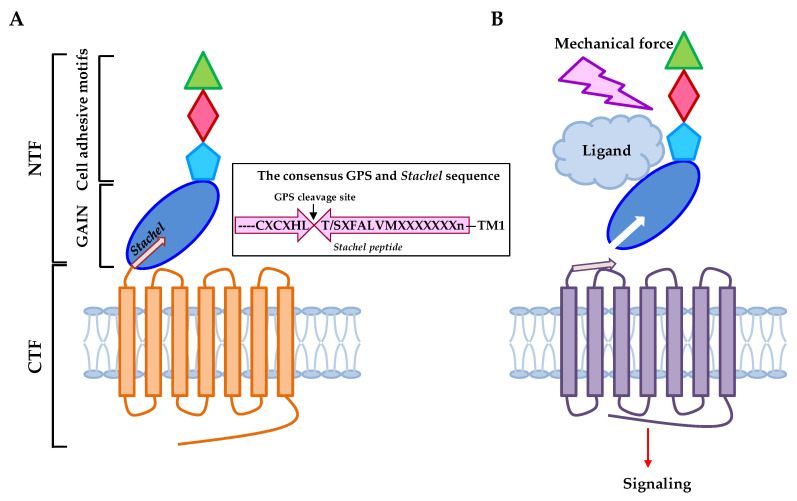
Structural characteristics and the tethered agonism model of aGPCRs. (**A**) The representative structural organization of aGPCRs in general [24]. The ECR contains N-terminal cell-adhesive protein motifs (indicated by colored shapes) followed by a GAIN domain in which the consensus GPS motif is located [28]. GPS autoproteolysis cleaves at the conserved HL*T/S sequence and generates the *Stachel* sequence (depicted as an arrow). CTF, C-terminal fragment; NTF, N-terminal fragment; GAIN, GPCR autoproteolysis-inducing. (**B**) The tethered agonist activation mechanism of aGPCRs [28,29,30]. Upon ligand binding and/or mechanical stimulation, the NTF is dissociated from the CTF to expose the *Stachel* peptide which binds to the 7TM region and induces receptor activation and signaling.

**Figure 2 pharmaceuticals-15-00219-f002:**
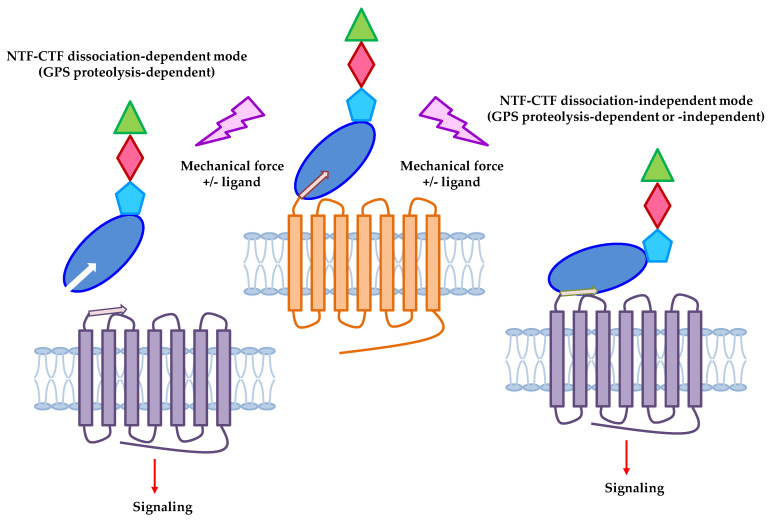
Potential activation mechanisms of mechanosensitive aGPCRs. Mechanosensitive aGPCRs might be activated by mechanical stimulation in the absence or presence of specific ligands via the NTF-CTF dissociation-dependent (**left**) or NTF-CTF dissociation-independent (**right**) mechanism [8]. In the dissociation-dependent mechanism, GPS proteolysis of aGPCRs and NTF shedding are absolutely required for the exposure of the *Stachel* peptide and the activation of CTF. In contrast, aGPCRs can be activated via a GPS proteolysis-dependent or -independent manner in the dissociation-independent mechanism. In this model, the *Stachel* peptide is partially exposed and bound to the 7TM region due to conformational changes of ECR induced by ligand/mechanical stimuli.

**Figure 3 pharmaceuticals-15-00219-f003:**
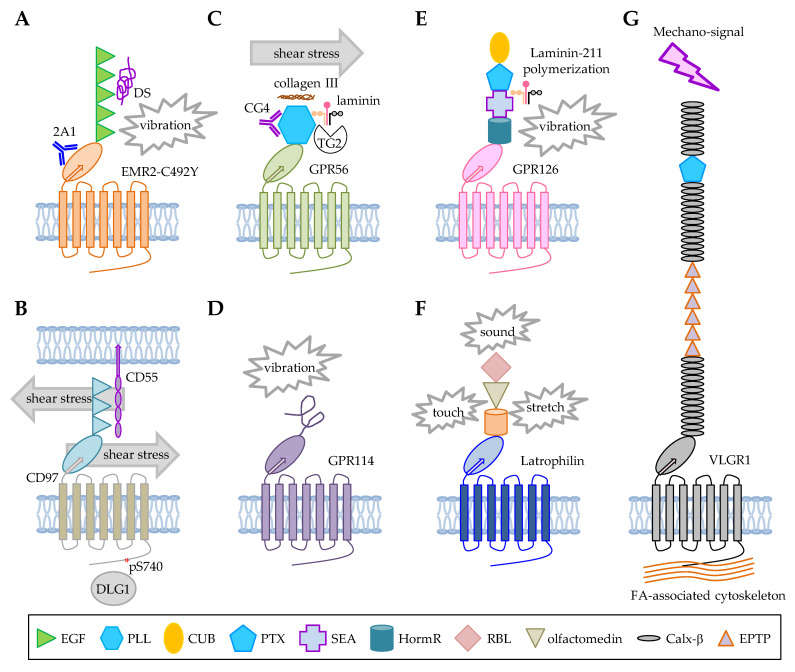
Summary of the structural features and activation mechanisms of mechanosensitive aGPCRs. (**A**) EMR2 activation in mast cells by vibratory shaking in the presence of DS or mAb [9]. (**B**) CD97 activation in leukocytes by CD55 and blood flow shear stress in vivo [10]. (**C**) GPR56 activation by collagen III plus blood flow shear stress in vivo [12] and by TG2/laminin in vitro [51]. (**D**) GPR114 activation by vibratory shaking in vitro [11]. (**E**) GPR126 activation by laminin-211 in the nervous system of zebrafish and by vibratory shaking in vitro [52]. (**F**) dCIRL/latrophilin activation by tactile, proprioceptive, and auditory stimuli in the nervous system of *Drosophila* larvae [14]. (**G**) VLGR1 activation by mechano-stimulation in vitro [13]. Various colored shapes listed in the lower panel represent the unique cell-adhesive protein motifs as indicated. The diagrams of receptors are not drawn to scale.

**Table 1 pharmaceuticals-15-00219-t001:** Summary of the molecular characteristics of mechanosensitive aGPCRs.

Receptor	Ligand	GPS Cleavage	Mechanical Force	In Vitro/In Vivo Evidence	References
ADGRE2/EMR2	DS, Ab	Dependent	Vibratory shaking	Vibratory urticaria, in vitro cell-based system	[9]
ADGRE5/CD97	CD55	Dependent	Shear stress	Blood flow/animal model	[10]
ADGRG1/GPR56	Collagen III TG2/laminin, Ab	Dependent Dependent	Shear stress N/A	Blood flow/animal model in vitro cell-based system	[12] [51]
ADGRG5/GPR114	N/A	Independent	Vibratory shaking	in vitro cell-based system	[11]
ADGRG6/GPR126	laminin-211	Dependent	Ligand polymerization	Zebrafish model, in vitro cell-based system	[52]
ADGRL1/LPHN1/CIRL	N/A	Independent	Touch, sound, stretch	Drosophila model	[14]
ADGRV1/VLGR1	N/A	N/A	mechanical stretch	in vitro cell-based system	[13]

## Data Availability

Not applicable.

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
