# Peer review of "Ligands and Beyond: Mechanosensitive Adhesion GPCRs"

_pharmaceuticals, 2022, doi:10.3390/ph15020219_

Round 1
Reviewer 1 Report
The Review article submitted to Pharmaceuticals by Hsi-Hsien Lin and colleagues provides a wide and interesting perspective on aGPCRs, their features and their role as mechanosensitive receptors.
The manuscript is informative, well prepared, and nice to read. Overall, it meets the standard for publication in the current journal. This Review is well written from Introduction throughout to Conclusions, and I report some minor observations below.
- Concerning Figures, artworks are clear, but the authors used “pale” colors that may be difficult to distinguish well in printed version. They could modify this. Moreover, I noticed that font is really small in some cases (for example in Figure 1 and, to some extent, in Figure 3).
- Besides Figures 1-3, the authors did not introduce any artwork in the following paragraphs. Artworks, such as schemes or, even more importantly, 3D structures of some of the key proteins/interactors (that can be retrieved from the Protein Data Bank), would greatly facilitate the reader. Another option would be to graphically represent sequence homology, when needed, this point is in fact only discussed within the text.
- Concerning Conclusions, the authors could discuss more about the “various physio-pathological significance” of the macromolecular assembly, for example by resuming some pathologies involved and, possibly, examples of therapeutical applications. In the same part, I suggest substituting “development of possible therapeutic reagents” with “development of possible therapeutic agents and drug candidates”.
Author Response
The Review article submitted to Pharmaceuticals by Hsi-Hsien Lin and colleagues provides a wide and interesting perspective on aGPCRs, their features and their role as mechanosensitive receptors.
The manuscript is informative, well prepared, and nice to read. Overall, it meets the standard for publication in the current journal. This Review is well written from Introduction throughout to Conclusions, and I report some minor observations below.
We thank the Reviewer for the positive comments.
- Concerning Figures, artworks are clear, but the authors used “pale” colors that may be difficult to distinguish well in printed version. They could modify this. Moreover, I noticed that font is really small in some cases (for example in Figure 1 and, to some extent, in Figure 3).
As suggested, we have modified the Figures and enlarged the font of the Figures.
- Besides Figures 1-3, the authors did not introduce any artwork in the following paragraphs. Artworks, such as schemes or, even more importantly, 3D structures of some of the key proteins/interactors (that can be retrieved from the Protein Data Bank), would greatly facilitate the reader. Another option would be to graphically represent sequence homology, when needed, this point is in fact only discussed within the text.
We thank the Reviewer for the comment. Due to the space limitation, we choose to use Figures 1-3 to represent the general features (structural characteristics and the tethered agonism model) of aGPCRs, the potential activation mechanisms of mechanosensitive aGPCRs, and the summary of distinct mechanosensitive aGPCRs, respectively. In our view, the inclusion of 3D structures of key ligand proteins/interactors of aGPCRs seems beyond the scope of the present review.
- Concerning Conclusions, the authors could discuss more about the “various physio-pathological significance” of the macromolecular assembly, for example by resuming some pathologies involved and, possibly, examples of therapeutical applications. In the same part, I suggest substituting “development of possible therapeutic reagents” with “development of possible therapeutic agents and drug candidates”.
We thank the Reviewer for the specific suggestion. We have added a couple of specific examples to highlight the “various physio-pathological significance” of mechanosensitive aGPCRs. We have also substituted the “development of possible therapeutic reagents” with “development of possible therapeutic agents and drug candidates”.
Reviewer 2 Report
In the Review, the authors discuss the possible mechanism of mechanotransduction of aGPCR (adhesion G protein-coupled receptors) based on their structural and functional characteristics. The undertaken problem seems to be an important issue to understand the mechanism of aGPCR activation, to identify their specific ligands and eventually find a more effective therapeutic approach for diseases affecting aGPCR signalling. The article message, however, should be provided clearer throughout the manuscript. In the current version, it is unclear whether the article's main goal is to convince readers to mechanotransduction of aGPCR, describe tethered ligands for aGPCR or just characterize aGPCRs. Authors should define the main goals of the Review Article precisely and reorganize the manuscript to clarify and convince the readers that the manuscript discusses these goals.
Specific comments:
1)Title of the manuscript is not specific. Please rephrase it to be more related to discussed in the Article issue.
2)section 2: structural characteristics and activation mechanisms including the „tethered agonism model”, “autoproteolysis” should be described shorter and clearer. Please correct.
3)section 3 together with subsections 3.1-3.7 seem to be unrelated to previously provided information in the introduction (section 1) and characteristics of aGPCR (section 2). Please complete chapter 3 with the explanation of how sections 1 and 2 are interconnected with subsections 3.1-3.7
4)Line 15: The phrase “mechano-sensors” is not precise. Please specify.
5)Line 18-19:
“Adhesion GPCRs (aGPCRs)”. Did you mean > Adhesion GPCR (aGPCR)<? I point out this because throughout the manuscript the abbreviation “aGPCR” has been used.
“the second largest GPCR subfamily with”. This information is confusing because you do not mention other than aGPCR, subfamilies of GPCR in the abstract. Please rephrase.
6)Since all ionotropic receptors are ion channels it is unclear what the authors mean by the phrase “channel-like ionotropic receptors”(Line 31). Please rephrase or specify channel-like receptors you characterize here.
7)lines 36-48: Characteristics of aGPCR is not complete, contains mental shortcuts of authors and seem to be unrelated to the rest of the article. Please improve it by providing the necessary knowledge to better understand the rest of the body text.
8)Line 41. Please add appropriate literature to the significance of aGPCR in “brain cortex development”; “fertility”. Also, please rephrase the sentence in line 40-42 to indicate precisely that cited review [11] do not “document” the functional significance of aGPCR but readers can find precise knowledge which is out of the scope of the present review.
9)General comment for review articles cited in the manuscript. Typically, Review Articles collect and confront conclusions from published original research to summarize a certain area of study. Sparing citation of review articles to provide information on where out-of-scope knowledge may be acquired is acceptable. However, in the current version of the manuscript, the authors cite a huge amount of review articles (37 which is 30% of all citations). Please, use appropriate references in the corrected version of the manuscript to convince readers that the article is a reliable summary of source research on the topic in question.
10)Line 55: did you mean >amino acid residues< instead of „residues”?
11)Please add references in Table 1 to data described in rows 2-8.
12) There is no information about the source of data presented in Figures 1-3. Please add appropriate references.
13)“more than one fifth of..” (Line 554). It is not clear how the authors calculated it. The authors do not give a precise explanation of this calculation in the body text. Please remove from the paragraph “Conclusions and future perspectives” information not discussed in the rest of the manuscript.
14)Line 556 phrase “aGPCR activation induction” is not clear. Please explain better the mechanism of the induction of the aGPCR activation or rephrase the sentence.
15) English grammar needs to be corrected by a proofreader specializing in scientific English before submitting it to Editor again.
Author Response
In the Review, the authors discuss the possible mechanism of mechanotransduction of aGPCR (adhesion G protein-coupled receptors) based on their structural and functional characteristics. The undertaken problem seems to be an important issue to understand the mechanism of aGPCR activation, to identify their specific ligands and eventually find a more effective therapeutic approach for diseases affecting aGPCR signalling. The article message, however, should be provided clearer throughout the manuscript. In the current version, it is unclear whether the article's main goal is to convince readers to mechanotransduction of aGPCR, describe tethered ligands for aGPCR or just characterize aGPCRs. Authors should define the main goals of the Review Article precisely and reorganize the manuscript to clarify and convince the readers that the manuscript discusses these goals.
We thank the reviewer for the suggestion. We have changed the last sentence of Abstract to “Here, we present a critical review of current evidence on mechanosensitive aGPCRs.” to focus on the main goal of discussing mechanosensitive aGPCRs. In addition, we have emphasized in the last part of Introduction that the main aim of the present Review is to summarize the current understandings of specific aGPCRs that function as potential mechanosensitive receptors. Furthermore, we have re-organized the manuscript as suggested by the Reviewer. The revision details are described below.
Specific comments:
Title of the manuscript is not specific. Please rephrase it to be more related to discussed in the Article issue.
We have changed the title of the Review article to “Ligands and Beyond: Mechanosensitive Adhesion GPCRs”
2)section 2: structural characteristics and activation mechanisms including the „tethered agonism model”, “autoproteolysis” should be described shorter and clearer. Please correct.
Please see the response below.
3)section 3 together with subsections 3.1-3.7 seem to be unrelated to previously provided information in the introduction (section 1) and characteristics of aGPCR (section 2). Please complete chapter 3 with the explanation of how sections 1 and 2 are interconnected with subsections 3.1-3.7
For comments 2 and 3, the following changes are made based on the suggestion of the Reviewer. We have made the best effort to discuss aGPCR autoproteolysis as concise as possible while keeping all essential information. We have removed “that occurs during aGPCR biosynthesis” from the sentence “The GAIN domain is an evolutionarily conserved protein configuration minimally required and self-sufficient to initiate a novel post-translational auto-proteolytic process that occurs during aGPCR biosynthesis” to further shorten the discussion.
We have rephrased the sentence “As described, autoproteolysis at the GPS motif would generate a new short N-terminal sequence (usually starting at Thr or Ser) of the CTF. Nevertheless, structural analysis of the GAIN domain has shown that this cryptic tethered peptide (also called the Stachel sequence) of CTF is usually embedded in and closely surrounded by the rest of the GAIN domain in the NTF” to “As described, autoproteolysis at the GPS motif would generate a new short N-terminal sequence (usually starting at Thr or Ser) of the CTF (also called the Stachel sequence), which is usually embedded in and closely surrounded by the rest of the GAIN domain”.
To better connect the Sections 1 and 2 with the Section 3, we have moved the original last paragraph of the Section2 to the beginning of the Section 3
4)Line 15: The phrase “mechano-sensors” is not precise. Please specify.
We have rephrased “mechano-sensors” as “mechanosensitive receptors” in the Abstract.
5)Line 18-19: “Adhesion GPCRs (aGPCRs)”. Did you mean > Adhesion GPCR (aGPCR)<? I point out this because throughout the manuscript the abbreviation “aGPCR” has been used.
We thank the reviewer for this comment. As there are 33 distinct human aGPCR members, we therefore used Adhesion GPCRs (aGPCRs) when all aGPCRs are indicated. In contrast, we used aGPCR when describing aGPCR in general.
“the second largest GPCR subfamily with”. This information is confusing because you do not mention other than aGPCR, subfamilies of GPCR in the abstract. Please rephrase.
We have added “According to the GRAFS classification system of GPCRs” before Adhesion GPCRs in the Abstract to highlight this unique GPCR subfamily.
6)Since all ionotropic receptors are ion channels it is unclear what the authors mean by the phrase “channel-like ionotropic receptors”(Line 31). Please rephrase or specify channel-like receptors you characterize here.
We have rephrased “ion channels and channel-like ionotropic receptors” to “ionotropic receptors”.
7)lines 36-48: Characteristics of aGPCR is not complete, contains mental shortcuts of authors and seem to be unrelated to the rest of the article. Please improve it by providing the necessary knowledge to better understand the rest of the body text.
Please see the response below.
8)Line 41. Please add appropriate literature to the significance of aGPCR in “brain cortex development”; “fertility”. Also, please rephrase the sentence in line 40-42 to indicate precisely that cited review [11] do not “document” the functional significance of aGPCR but readers can find precise knowledge which is out of the scope of the present review.
For comments 7 and 8, we have added new references to denote the relevant original research articles for the specific functions of aGPCRs such as cortex development and fertility mentioned in the text. We have also added a phrase to indicate that the functional significance of aGPCRs is out of the scope of the present review. For the characteristics of aGPCRs mentioned in the Introduction section, we highlighted briefly the classification of aGPCR subfamily, as well as the functional significance, structural-functional relationship, ligand identification, and possible activation mechanisms of aGPCRs to pave the way for the 2nd section: Adhesion GPCRs: structural characteristics and activation mechanisms. While not comprehensive, these are relevant for the discussion of aGPCRs as mechanosensitive receptors.
9)General comment for review articles cited in the manuscript. Typically, Review Articles collect and confront conclusions from published original research to summarize a certain area of study. Sparing citation of review articles to provide information on where out-of-scope knowledge may be acquired is acceptable. However, in the current version of the manuscript, the authors cite a huge amount of review articles (37 which is 30% of all citations). Please, use appropriate references in the corrected version of the manuscript to convince readers that the article is a reliable summary of source research on the topic in question.
We thank the comment of the Reviewer. We have now added many relevant original research articles in the reference list of the Sections 1-3 to the best of our ability.
10)Line 55: did you mean >amino acid residues< instead of „residues”?
We have changed “residues” to “amino acids”.
11)Please add references in Table 1 to data described in rows 2-8.
References have been listed as suggested.
12) There is no information about the source of data presented in Figures 1-3. Please add appropriate references.
References have been listed in the figure legends as suggested.
13)“more than one fifth of..” (Line 554). It is not clear how the authors calculated it. The authors do not give a precise explanation of this calculation in the body text. Please remove from the paragraph “Conclusions and future perspectives” information not discussed in the rest of the manuscript.
We have added (7 out of 33) in the sentence to explain the reason for the claim “more than one fifth of”. As discussed in the main text of the Review article, 7 distinct aGPCRs are potential mechano-sensitive receptors. There are a total of 33 different aGPCRs and 7/33 means more than one fifth of aGPCRs.
14)Line 556 phrase “aGPCR activation induction” is not clear. Please explain better the mechanism of the induction of the aGPCR activation or rephrase the sentence.
We have rephrased “aGPCR activation induction” to “aGPCR activation mechanism”.
15) English grammar needs to be corrected by a proofreader specializing in scientific English before submitting it to Editor again.
We have asked colleagues who are English natives to proofread the manuscript for correct grammar.
Round 2
Reviewer 2 Report
The Article revised by the authors is significantly corrected and improved. The authors applied suggested corrections efficiently. I have no additional comment. In my opinion the Article, Ligands and Beyond: Mechanosensitive Adhesion GPCRs by Hsi-Hsien Lin H-H et al. is ready for publication in Pharmaceuticals.